# Aluminum Electrodeposition on the Surface of Boron Carbide Ceramics by Use EMIC–AlCl₃ Ions Liquid

Roujia Gou [1,*], Jae-Hyeok Park [2], Seiji Yamashita [3], Takeshi Hagio [1,2], Ryoichi Ichino [1,2] and Hideki Kita [1]

1. Department of Chemical Systems Engineering, Graduate School of Engineering, Nagoya University, Furo-cho, Chikusa-ku, Nagoya 464-8603, Japan
2. Institute of Materials Innovation, Institutes of Innovation for Future Society, Nagoya University, Furo-cho, Chikusa-ku, Nagoya 464-8601, Japan
3. Department of Materials Process Engineering, Graduate School of Engineering, Nagoya University, Furo-cho, Chikusa-ku, Nagoya 464-8603, Japan
* Correspondence: gou.roujia@f.mbox.nagoya-u.ac.jp

**Abstract:** Coating technology is decisively important for metallization of ceramic materials and ceramic metal sealing technology. Previous studies have shown that the network-like structure after penetration of molten aluminum can significantly improve the strength of joint components. However, the direct aluminum coating method is limited by the shape of the substrate. To obtain a dense aluminum film on the surface of $B_4C$, in this study, aluminum was deposited by pulse electroplating in EMIC–AlCl₃ ionic liquid. The deposited metals were observed and analyzed by SEM–EDS and XRD. A Vickers hardness tester was adopted as an auxiliary equipment to clarify the film quality. The results show that frequency and duty cycle have significant effects on crystal orientation. The content of oxides in the contact gap reduces the bonding strength of the deposited metal, which provides experimental basis for metal electrodeposition on $B_4C$.

**Keywords:** $B_4C$ ceramics; EMIC-AlCl₃; electrodeposition; aluminum

## 1. Introduction

Boron carbide ($B_4C$) has the advantages of a high melting point, high hardness, low density, good wear resistance, strong acid and alkali resistance, and high electrical conductivity [1]. Hence, it has been used in various fields including in chemical industries and machinery. However, because of its low fracture toughness, the damage caused during processing and production decreases the material strength to below its ideal value. Moreover, under normal circumstances, the firing temperature of $B_4C$ ceramics is approximately 2200 °C, the energy consumption for firing is high, and it cannot be recycled after being damaged. Therefore, if the fracture toughness of the ceramic material can be improved or its surface can be restored, it can not only extend the service life of ceramics, but also save energy and protect the environment. Metals have excellent electrical and thermal conductivities and ductility and are easy to process. Previous studies [2–6] have shown that a combination of metals and ceramics can yield composite materials with better performance.

Sekine et al. [7] showed that at a temperature of 700 °C, an infiltration phenomenon occurs in the gap between the metal Al and the $B_4C$ joints, and the joint strength is greater than that of the base material itself. Osada et al. [8] also mentioned the use of metal fluidity at high temperatures to heal ceramic materials. In the bonding process, an aluminum (Al) foil is typically used as the bonding material. Although the direct lamination method is simple to operate, this method performs better on parts with simple shapes. For components with complex structures, the bonding performance of the direct coating method is not ideal. The surface metallization treatment of $B_4C$ with metal Al and the method of heating to make the metal penetrate cracks may help cure the surface cracks.

Metallization at the ceramic surface is important for dependable joints of ceramic metals. However, different crystal lattices and types of bonds lead to low wettability. In addition, a significant difference in the coefficient of thermal expansion and melting point temperature makes effective bonding between ceramics and metals difficult.

The possible materials for metallization are limited. Chmielewski et al. [9] proposed a process experiment to prepare Mo–$Al_2O_3$ composites by hot pressing to overcome brittleness, which is a major technical limitation in the wide application of ceramic materials. Lee et al. [10] used the surface modification technology to braze $Al_2O_3$–SUS304 with a conventional Ag–Cu eutectic solder. Olesińska et al. [11] described the results of the formation of a barrier layer when copper was attached to copper on AlN ceramics using the copper direct connection technique. The second, more common method, is to create a thin metal coating on the ceramic surface, which can be attached to the metal relatively easily using conventional welding methods. Physical deposition methods, such as plasma spray [12], physical or chemical vapor deposition [13], electroless plating [14], and hot dipping [15], are often used to obtain metal coatings on ceramic surfaces. However, compared with these methods, the electrodeposition method has the advantages of no need heating and better adaptability for complex shape components when deposition, simple operation, and good control of the sedimentary layer structure, and it has been widely used in industries. There has been significant research in recent years, and this method has been improved. Al and its alloys are promising materials because of their low density, high corrosion resistance, and high electrical conductivity. Due to the active chemical properties of Al, the precipitate potential is lower than that of hydrogen ions, and Al cannot be precipitated successfully in aqueous electrolytes. A mixture of 1-ethyl-3-methyl-imidazolium chloride (EMIC) and aluminum chloride ($AlCl_3$) has been used as an ionic liquid for the electrodeposition of Al, using with various metals and alloys, such as Al [16], Al–Zn [17], Al–Mo–Ti [18], and even AlInSb semiconductors [19], have been electrodeposited.

Pulsed current technology has been used in the electrodeposition of Al [20,21] because it can be controlled and improved by adjusting the pulse parameters such as the frequency, duty cycle, pulse strength, and polarity. Yang et al. [20] studied the effects of electrolyte composition, current density, and current form on the characteristics of the Al layer deposited by the $AlCl_3$-n-butylpyridine chloride molten salt system. The authors found that the Al layer deposited by the pulsed current method was denser and smoother than that deposited using the direct current method. Li et al. [21] investigated the influence of pulse parameters on the deposition of Al by an $AlCl_3$–EMIC ionic liquid containing $NdCl_3$. They showed that the micro-deposition changed from polygonal crystals to spherical crystals with increasing DC current and that the pulsed current made the micro-deposition brighter and flatter than that under the DC method at the same average current density.

Kita et al. [7,22] showed that the penetration of pure Al film with the direct coating method into cracks improves the mechanical properties of the material. In our previous study [23], the method employed by Kita et al. was used to obtain coatings; however, it was limited because of some complicated parts. Electroplating, by contrast, is shape-free and easier to control. Although $B_4C$ is known to exhibit conductivity, no study has described the formation of an Al film by electroplating using these properties. Therefore, based on industrial requirements, we investigated electrodeposition coating and deposited Al on $B_4C$ using the EMIC–$AlCl_3$ ionic liquid.

The purpose of this study was to determine the optimum process conditions for obtaining a dense film of Al on the surface of $B_4C$ by electroplating and to clarify the film quality under varying conditions.

## 2. Materials and Methods

### 2.1. Basic Parameters of $B_4C$

$B_4C$ (provided by Mino Ceramics Co., Ltd., Tokyo, Janpan) was used as the substrate. Substrate samples were cut into cuboids with dimensions of 3 mm × 4 mm × 7 mm and

mechanically polished in air. Subsequently, they were ultrasonically degreased in ethanol and acetone for 15 min. Table 1 lists its basic parameters.

**Table 1.** Basic parameters of $B_4C$ tested.

| Density (g/cm$^3$) | Young's Modulus (GPa) | Four-Point Bending Stress (MPa) | Electrical Conductivity (at 25 °C) (S/m) |
|---|---|---|---|
| 2.45 | 430 | 240 ± 14 | 140 [24] |

*2.2. Ionic Liquid Preparation and Electrodeposition*

An acidic EMIC–AlCl$_3$ ionic liquid was prepared using EMIC (97%, Tokyo Chemical Industry Co., Ltd., Tokyo, Japan) and AlCl$_3$ (99.99%, Kojundo Chemical Lab. Co., Ltd., Sakado, Japan) in a glass beaker and mixed with a molar ratio of 1:2 at room temperature. The mixture was stirred for over 1 h to ensure homogeneity. An Al wire (99.99%, The Nilaco Corporation, Tokyo, Japan) was soaked in prepared EMIC–AlCl$_3$ ionic liquid for more than one week to remove trace amounts of water remaining in the ionic liquid.

B$_4$C substrates were priorly rinsed by ultrasonication in distilled water, 0.1 M sodium hydroxide solution, and acetone, respectively, to remove surface impurities, such as grease. In the electroplating experiment, a pure Al wire was used as the anode and Al rod was used as the reference electrode. The electrodeposition area of B$_4$C was fixed at 2.0 × 0.3 cm$^2$, and the extra area was masked with insulation tape. The ambient temperature was 25 °C. Figure 1 shows a schematic diagram of the Al electroplating process using EMIC–AlCl$_3$ ionic liquid. The relevant chemical reactions are as follows [25,26]:

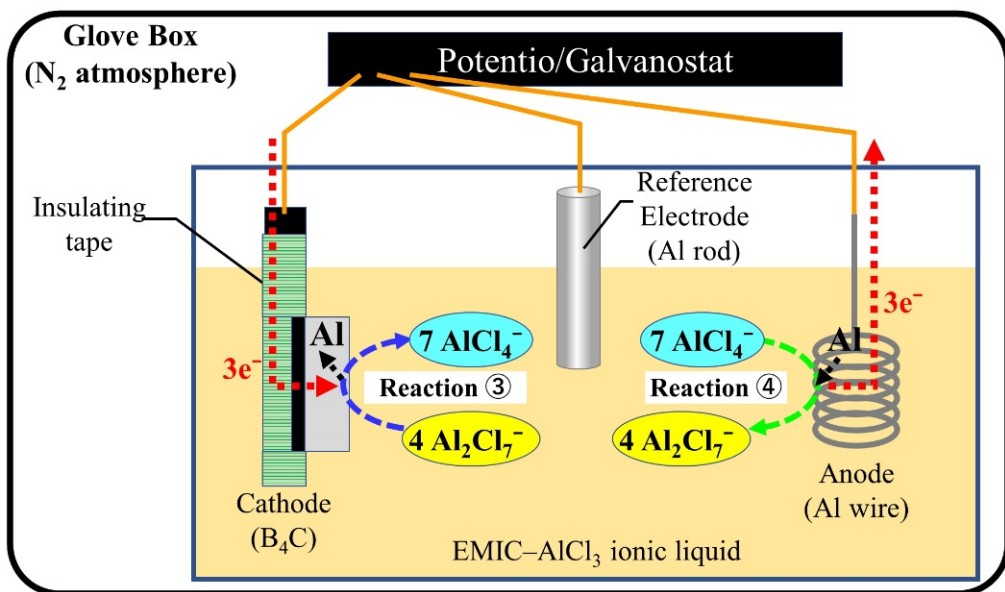

**Figure 1.** Schematic diagram of the Al electroplating process using EMIC–AlCl$_3$ ionic liquid.

①　Ionic liquid: $AlCl_3 + EMIC \rightarrow AlCl_4^- + EMI^+$
②　Internal reaction: $AlCl_4^- + AlCl_3 \rightarrow Al_2Cl_7^-$
③　Cathode reaction (reduction): $4Al_2Cl_7^- + 3e^- \rightarrow Al(s) + 7AlCl_4^-$
④　Anode reaction (oxidation): $Al(s) + 7AlCl_4^- \rightarrow 4Al_2Cl_7^- + 3e^-$

Protons have a particularly stronger ability to bind with electrons rather than Al ions; thus, the inclusion of water causes hydrogen generation and inhibits Al deposition reaction in the ionic liquid. Moreover, Al has an active chemical nature, making it easy to react with oxygen to form alumina. To impair the effect of these factors, all experiments were conducted in a glove box filled with N$_2$. In addition, the uniform coating cannot form on boron carbide ceramics by constant current. It was due to this that the pulse current

was used for this experiment. There are three parameters for controlling the deposition, which are current density, frequency, and duty ratio, to obtain the uniform formation. The experiments were conducted to confirm the effects in different variate. Table 2 presents the specific experimental conditions when applying a constant charge of 60 C·cm$^{-2}$ (theorical plating thickness: approximately 20 μm).

**Table 2.** Experimental conditions.

| Sample | Current Density (mA/cm$^2$) | Frequency (Hz) | Duty Ratio |
|---|---|---|---|
| dAl-1 | −15 | 100 | 0.8 (8 ms-on/2 ms-off) |
| dAl-2 | −30 | 100 | 0.8 (8 ms-on/2 ms-off) |
| dAl-3 | −50 | 100 | 0.8 (8 ms-on/2 ms-off) |
| dAl-4 | −30 | 10 | 0.8 (80 ms-on/20 ms-off) |
| dAl-5 | −30 | 50 | 0.8 (16 ms-on/4 ms-off) |
| dAl-6 | −30 | 50 | 0.5 (10 ms-on/10 ms-off) |
| dAl-7 | −30 | 50 | 0.2 (4 ms-on/16 ms-off) |

　　　The dAl-1~3: effect of current density (−10, −30, −50 mA /cm$^2$). The surface cover density of coating judged by SEM, then choose the −30 mA/cm$^2$ as an in-variant current for the next step. The dAl-2, 4, 5: effect of frequency (current density fixed at −30 mA/cm$^2$, frequency is 100, 10, 50 Hz). Cover density and particle size of coating was judged by SEM observation, then choose the 50 Hz as an invariant frequency for the next step. The dAl-5, 6, 7: effect of duty ratio (fixed current density at −30 mA/cm$^2$, fixed frequency at 50 Hz, duty ratio 0.8, 0.5, 0.2).

*2.3. Analysis Method*

　　　The microstructures of the surfaces and cross sections and the elemental analysis of the deposits were observed using a field emission scanning electron microscope (FE-SEM, JEOL, JSM-7500F, Tokyo, Japan). The phase composition was determined by X-ray diffraction (XRD, Rigaku SmartLab, Tokyo, Japan) with Cu Kα radiation (λ = 1.5418 Å). The operating parameters were 40 kV and 30 mA, with a 2θ step size of 0.01. A Vickers hardness tester (FM-300e, FUTURE-TECH CORP, Kanagawa, Japan) was used to apply force and judge the quality of the deposition by changing the surface morphology. In order to reach the substrate, the load for judging the quality of deposition was 5 kg with holding time of 10 s. The load for measurement of the surface hardness by Vickers hardness tester was 0.1 kg.

## 3. Results

*3.1. Observation Results*

　　　Figure 2 shows the SEM observation images after electrodeposition. The appearance of the Al film after deposition was silvery white. The Al particle sizes in samples dAl-2, dAl-3, and dAl-5 were smaller than those in the samples dAl-1, dAl-4, dAl-6, and dAl-7, which presented agglomerated and spherical shapes, as shown in the observation results. In addition, the dAl-1, dAl-4, and dAl-7 present poor surface coverage and fewer deposition particle connections. The coating on dAl-7 has a slight peeling, indicating a poor adhesion between the coating and the substrate material.

　　　The dAl-1, dAl-2, and dAl-3 exhibit deposition morphology changes under different current densities (100 Hz, duty ratio of 0.8). When compared with dAl-5, dAl-6, and dAl-7, the SEM observation results show that when the current density and frequency are the same and the duty ratio is 0.5, larger crystals are obtained; however, the Al film deposited is not dense enough.

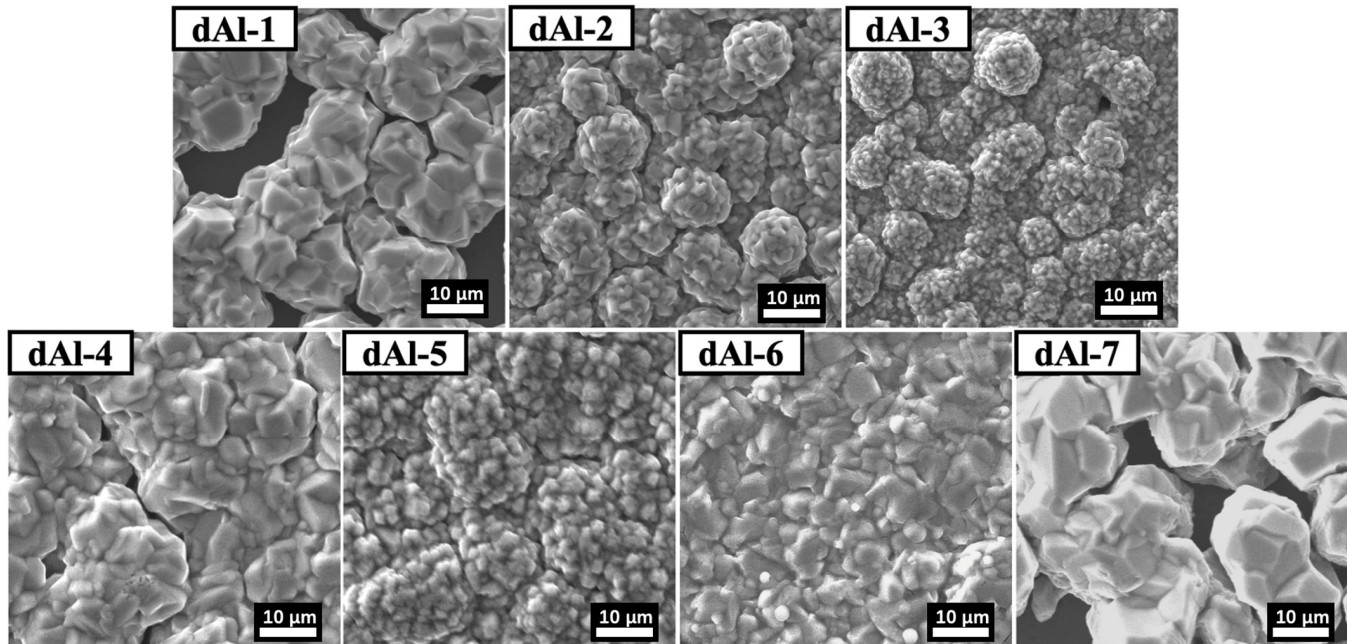

**Figure 2.** Surface morphologies of Al coatings on B$_4$C from EMIC–AlCl$_3$ ionic liquid under the different pre-treatments.

*3.2. XRD Analysis Results*

Figure 3 shows the XRD patterns of each sample. The (111), (200), (220), (311), and (222) surface characteristic peaks of Al can be detected. Metal Al is a face-centered cubic lattice structure which the (111) plane is the close-packed plane. With an increase in the current density, the intensity of the peak (111) decreases. This indicates that at a current density of $-15 \, \text{mA/cm}^2$, the area of (111) surface was larger. The difference in the diffraction peak heights between dAl-1 and dAl-2 is small, but compared with dAl-3, this difference is significant. The Al peak value of dAl-5 was the highest among the XRD patterns of dAl-2, dAl-4, and dAl-5. With the same plating time, the grain growth increased at a frequency of 10 Hz, which also increase the Al coverage rate.

The peak intensity of (111) for dAl-6 is the same as that of dAl-7. However, the (200) surface had the highest intensity in the dAl-7 sample. Based on the change in the intensity ratios of dAl-2, dAl-4, and dAl-5, the (200) surface intensity increased when the frequency was increased to 50 Hz. Therefore, we suggest that the crystal orientation changes with the frequency.

The crystal size was calculated using the Scherrer equation [27]. The detailed calculation results are presented in Table 3. The average grain size of the Al coatings was calculated from XRD patterns combined with the Scherrer equation given by [28,29]:

$$FWHM = (k \times \lambda)/(L \times cos \, \theta)$$

in which *FWHM* is the full width at half-maximum of the diffraction peak, *k* is a Scherrer's constant, *L* is the crystallite size, and *θ* is the Bragg angle.

To have the same thickness, the treatment times were different. The thickness of each film was measured by SEM observation and estimates. Compared with dAl-1, dAl-2, and dAl-3, which were deposited under different current densities, in Figure 2 and Table 2, when current densities decreased, the crystal size grew. Since the amount of current affects nucleation and growth, the higher the current density, the faster the nucleation speed and the smaller the particle size. At low current density, the duty ratio was unchanged, and the on-off time was the same, resulting in the growth rate of the crystal being greater than the nucleation rate. This shows that at low current densities, the size of the crystals and

the size of the deposited particles grow. In addition, a linear relationship appears when particle size is plotted versus inverse over potential. Furthermore, within each current density stratum, the Al particle size increases as the amount of Al deposited is increased. This effect is most pronounced with low current density deposition due to the relatively low particle density which causes individual particles to grow significantly with increasing Al capacity [25,30]. As shown in Table 3, the grain size of dAl-5 was the smallest with a duty ratio of 0.8. In addition, the growth rate of the surface of the sedimentary layer increases, and the greater spherical protrusions have made the surface rougher. There were nonsignificant effects under frequency changes as shown in the results of dAl-2, dAl-4, and dAl-5, but a high frequency reduced the grain size. With the increase in the frequency, the current duration is shortened, and the surface levelling is improved. However, when the frequency is increased to a certain extent, the interval to change the time-on of the current is too short to supplement metal ions to the cathode, and, at the same time, the hindering effect of the adsorbed substances on the effective interference of grain growth is reduced. Thus, the growth rate is greater than the nucleation rate, and the grain size is increased [31].

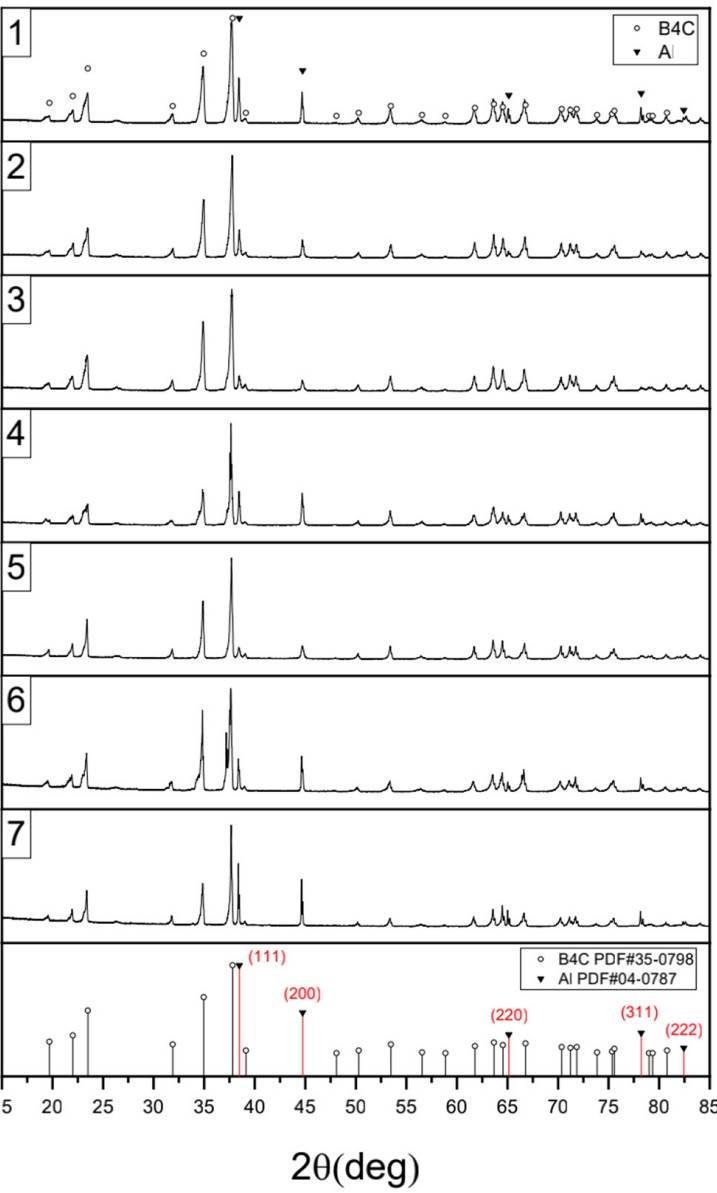

**Figure 3.** XRD analyses of samples in Table 2.

**Table 3.** Vickers hardness, plating thickness, and XRD analysis results.

| Sample | Hv | Average Size (nm) | Intensity Ratio (111/200) | Thickness (μm) |
|---|---|---|---|---|
| dAl-1 | 45.1 | 63 ± 2 | 1.49 | 20 ± 0.28 |
| dAl-2 | 95.3 | 39 ± 1 | 1.55 | 22 ± 0.78 |
| dAl-3 | 93.5 | 35 ± 1 | 1.41 | 19 ± 0.56 |
| dAl-4 | 46.5 | 56 ± 1 | 1.01 | 19 ± 0.37 |
| dAl-5 | 90.6 | 34 ± 2 | 0.89 | 20 ± 0.25 |
| dAl-6 | 47.4 | 59 ± 3 | 1.11 | 18 ± 0.91 |
| dAl-7 | 42.6 | 62 ± 1 | 1.47 | 18 ± 0.80 |

Based on the observation results of the dAl-5, dAl-6, and dAl-7, the reduced duty cycle and long disconnect time increases the time for grain growth. Therefore, the SEM observation results show that the sizes of the dAl-6 and dAl-7 are larger than those of dAl-5. The calculated grain size results are also consistent with the observed results. Moreover, from the intensity ratios of (111) and (200), it can be found that with an increase in the duty ratio, the preferred orientation of (111) decreases, while the preferred orientation of (200) increases.

### 3.3. Observation Results of Indentation Treatment

A Vickers hardness tester was used for detecting film bonding condition; the indentation was desired to run through the coating. For this reason, we performed on the coating side with a load of 5 kg and a pressure holding time of 10 s. The indentation did not treat the surfaces of dAl-1, dAl-4, and dAl-7 because of their low surface coverage. From the SEM observation results shown in Figure 4, fractures can be seen near the indentation area in dAl-3 and dAl-5, which are not observed in the other samples. However, the film increased around the indentation area of dAl-2 after loading, which is similar to the results of dAl-3 and dAl-5. The most destroyed face appeared in dAl-5, and the crack was the longest. The complete indentation morphology of sample dAl-6 was obtained, and no ridges or fracture defects were produced. This indirectly proves that the performance of the surface coating meets the requirements.

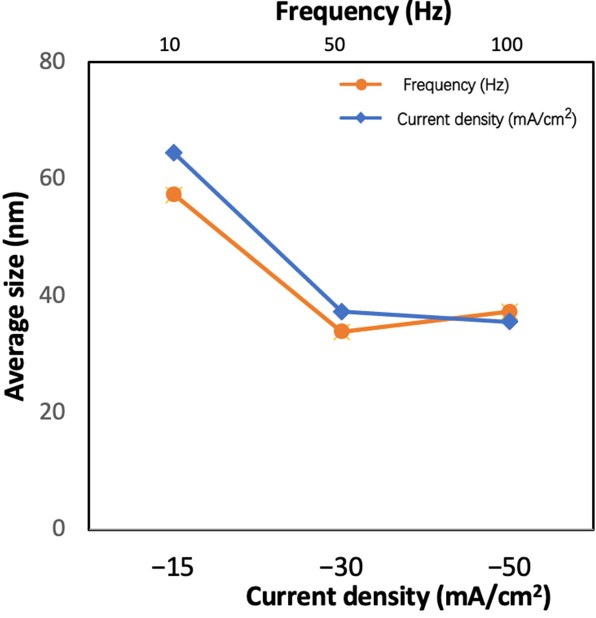

**Figure 4.** The relation between current density/frequency and an average particle size.

In addition, we compared the indentation position with the deposition surface observation map. As shown in enlarged observation of (c-1) and (d-1), the grains of dAl-5 are clearly separated, and the indentation position can be clearly seen even after plastic deformation under stress. Therefore, compared to dAl-6, dAl-5 has a higher brittle phase content, which affects the toughness strength of the deposited Al and leads to the generation of cracks. From the observation images of the cracks shown in Figure 5, we suggest that an intergranular fracture occurred at the deposited surface after loading. In order to observe the cross-section surface of coating, the three-point bending tester was used to break the sample in two pieces. After testing, the Al film of dAl-6 did not break into two pieces but peeled from the surface of B$_4$C. Figure 6 shows the cross-section SEM observation after the 3-point bending test. In Figure 6, it can be observed that the (a) and (b) coatings are arching on the surface of B$_4$C due to the external force; (c) and (d) did not escape from the substrate because the deposition particles gap released the force. In addition, we observed the metal fracture morphology of dAl-2, which had no cracks after the indentation treatment, and dAl-5, which had more surface cracks. The results, presented in Figure 7a–c represent the vertical direction of the observed crack and morphology of dAl-5. Figure 7d,e show the cross-sectional observation results of dAl-5 and dAl-2, respectively. Comparing the EDS mapping images of (d-1) and (e-1) in Figure 7, the oxygen distribution is different. In (e-1), the brittle oxide phase is deposited at the grain boundaries, which prevents the plastic relaxation and has hardening effects. The surface hardness value of smaller grains is higher, and the brittle phase between the grains (such as Al$_2$O$_3$) under the action of the external force appeared to have separated and fractured due to plastic deformation pulling, resulting in cracks [32,33].

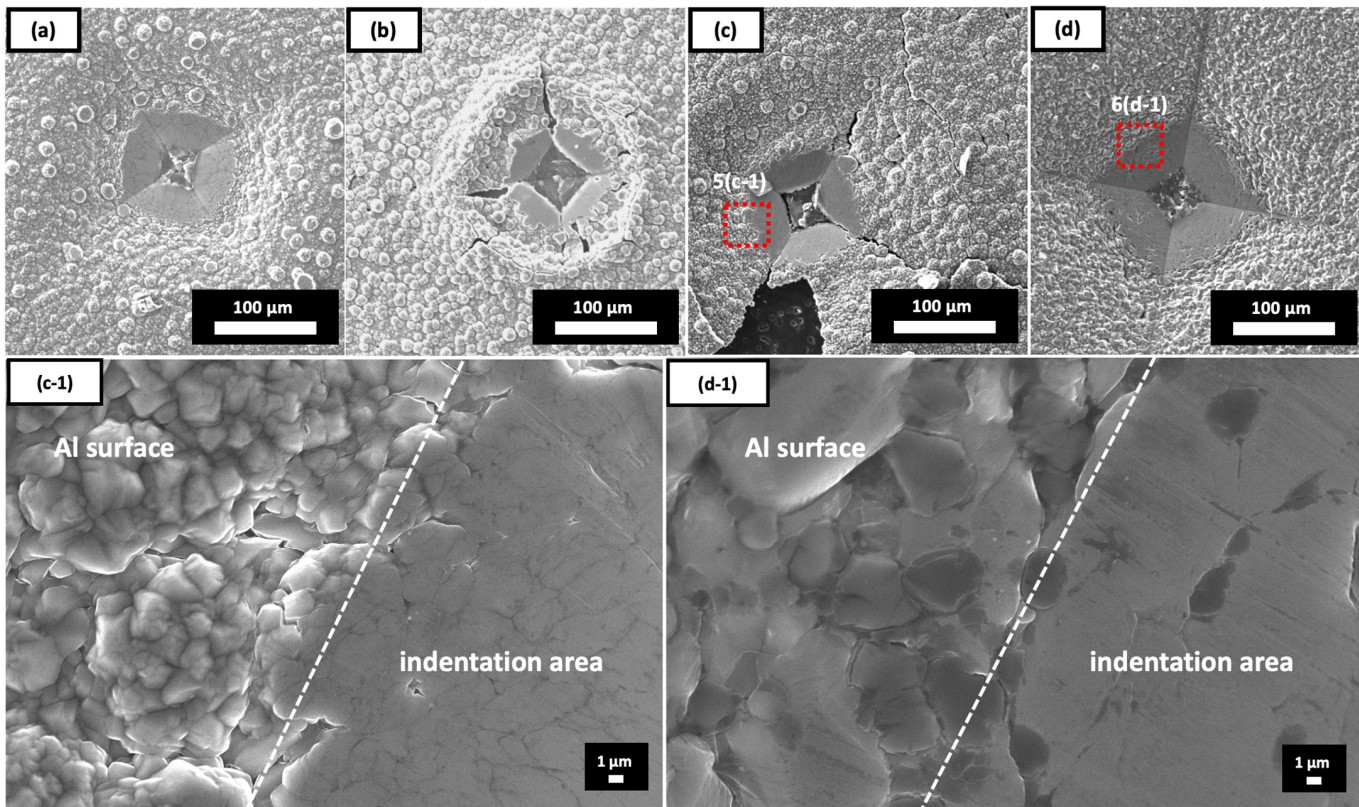

**Figure 5.** Indentation morphologies observations: (**a**): dAl-2; (**b**): dAl-3; (**c**): dAl-5; (**d**): dAl-6 by SEM; (**c-1**) and (**d-1**) are the enlarged observation of (**c**) and (**d**) separately.

**Figure 6.** The cross-section SEM observation after 3-point bending test, (**a**) dAl-2; (**b**) dAl-5; (**c**) dAl-7; (**d**) dAl-4.

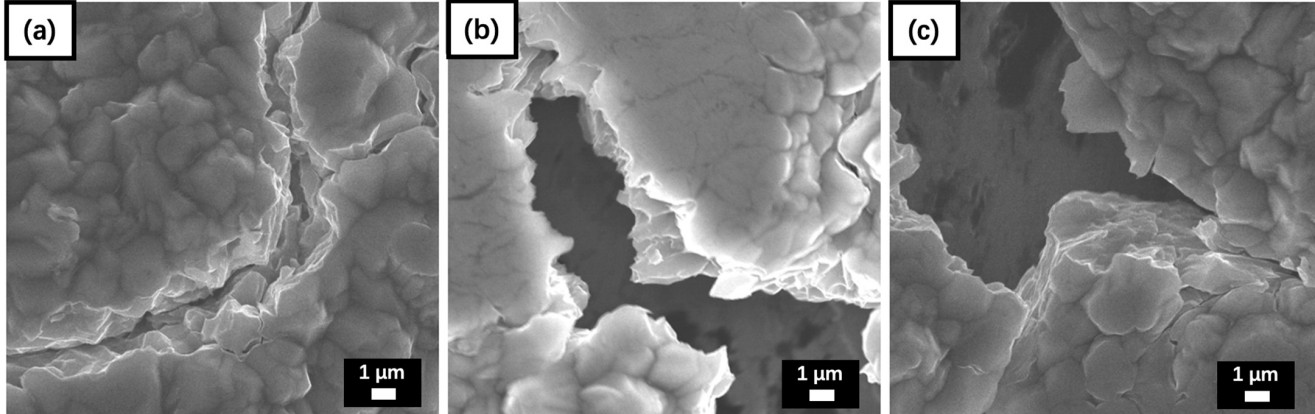

**Figure 7.** *Cont.*

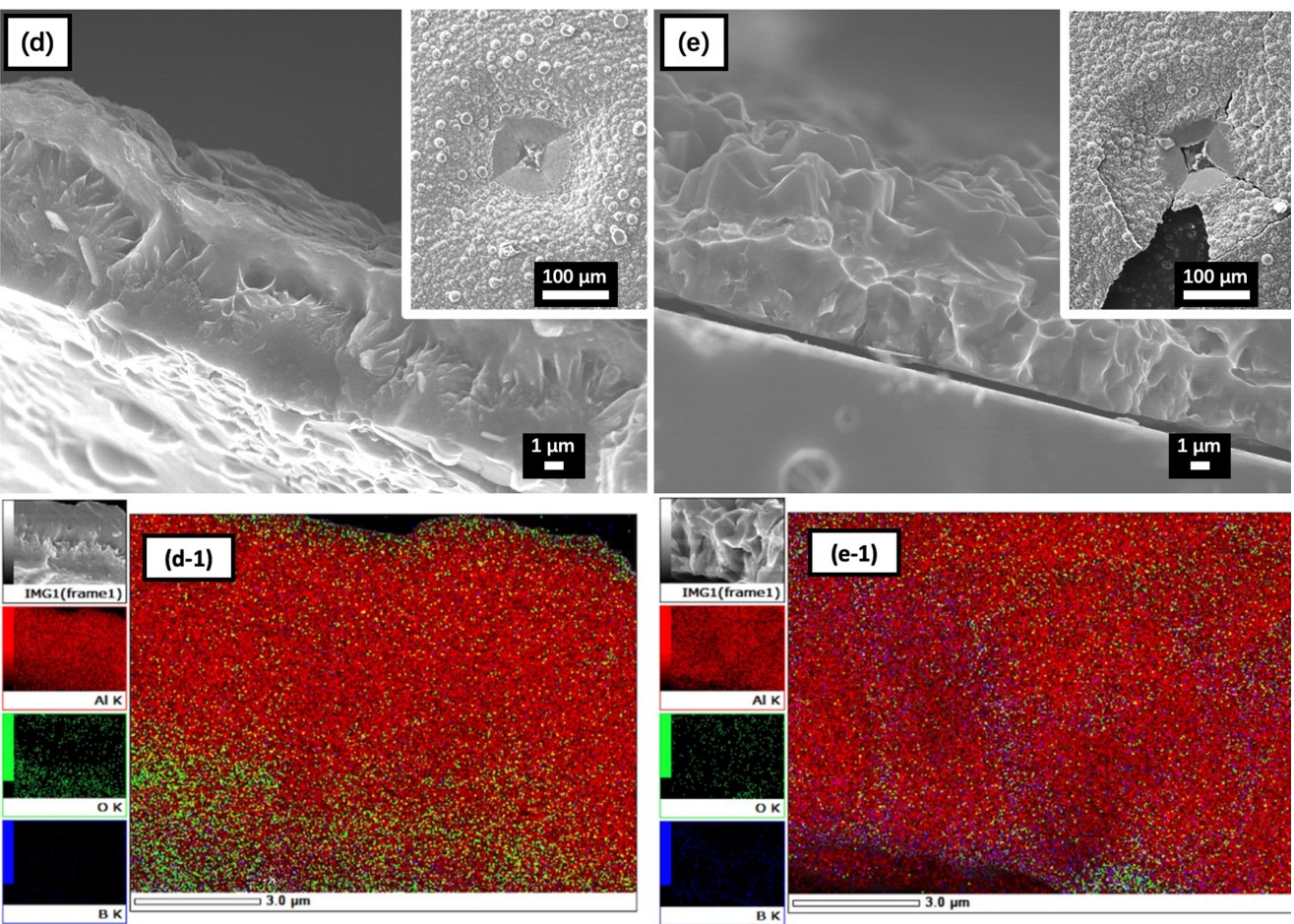

**Figure 7.** Observation results of cracks in sample dAl-5 are (**a–c**); (**d,e**) are cross-sectional observation results of dAl-2 and dAl-5, respectively; (**d-1**) and (**e-1**) EDS mapping images of cross-sectional observation.

Observations of the cracks in Figure 7 show that there are two types of fracture marks: the grain boundary fracture and the deformation tearing. The crystals of the Al on the fractured surface were relatively small and granular. From the observation results that the specific surface area of the deposited Al with a smaller particle size is higher; therefore, the Al film is oxidized in the gap as a brittle phase, which increases the hardness of the deposited Al on the surface. In addition, differences in elemental distribution can be seen by EDS surface scans. The analysis of the samples in combination with the morphology map reveals that the distribution of alumina is more pronounced in the fracture. Indirectly, it has proved that the secondary phase oxide has an influence on the quality of the deposited Al.

## 4. Conclusions

The grain size decreased with increasing deposition current density. In addition, the enhanced concentration polarization decelerated the deposition rate and decreased the thickness of the deposited film. The alumina film formed in the gap as a brittle phase, which increases the hardness of the deposited Al, but the oxides in the contact gap reduces the bonding strength of the deposited particles.

With the increase in the frequency, the current duration is shortened, and the surface levelling is improved. However, when the frequency is increased to a certain extent, the interval to charge the time-on of current is too short to supplement metal ions to the cathode, and, additionally, the hindering effect of the adsorbed substances on the effective

interference of grain growth is reduced. Thus, the growth rate is greater than the nucleation rate, and the grain size is increased. At low frequencies, increasing the duty cycle increased the time for grain growth. The duty ratio had a significant influence on the preferred orientation. Under a duty ratio of 0.5, the preferred orientation level of (200) was higher than that of (111).

In this study, a Vickers hardness tester was used to apply force and judge the quality of the deposition by changing the surface morphology. The dAl-6 ($-30$ mA, 50 Hz, 0.5) indentation observation shows no cracking appearance, which indirectly reflects the great bonding strength of deposition particles. The Vickers hardness tester practicable stands a chance of being a useful adjunct to judge the quality of the deposited film.

**Author Contributions:** Conceptualization, R.G. and H.K.; methodology, J.-H.P., T.H., R.I., R.G., S.Y. and H.K.; validation, T.H., S.Y. and R.G.; formal analysis, J.-H.P. and R.G.; investigation, J.-H.P., T.H., R.G. and S.Y.; resources, R.I. and H.K.; writing—original draft preparation, R.G.; writing—review and editing, J.-H.P., T.H., R.I., H.K., S.Y. and R.G.; supervision, R.I. and H.K.; funding acquisition, H.K. All authors have read and agreed to the published version of the manuscript.

**Funding:** This work has been financially supported by Japan Science and Technology Agency (JST)—Program on Open Innovation Platform with Enterprises, Research Institute and Academia (OPERA), Grant JPMJOP1843.

**Institutional Review Board Statement:** Not applicable.

**Informed Consent Statement:** Not applicable.

**Data Availability Statement:** Data is contained within the article.

**Conflicts of Interest:** The authors declare no conflict of interest.

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
