# Peer review of "Aluminum Electrodeposition on the Surface of Boron Carbide Ceramics by Use EMIC–AlCl3 Ions Liquid"

_coatings, doi:10.3390/coatings12101535_

Round 1

Reviewer 1 Report

Dear Authors: 

first of all, I congratulate you on the remarkable effort you have put into this work. 

In general, it seems to me an interesting research that aims to contribute to the knowledge of the formation of metallic films in electrochemical systems. However, I consider that the information precisely on the experimental system is very limited and you only detail the results very briefly.  I also consider that you should deepen the description of the electrochemical system to evaluate the potential of the technology. 

Here are some comments you might consider to strengthen your work: 

- In the title, please consider changing Ions Liquid to Ionic Liquid.

- Line 10: please check the sentence: Coating technology is a very important... I consider that the "a" should be omitted or use other words like crucial or decisive or determining, etc. 

-Line 47: please check the sentence: ionic liquid molten salt... liquid and molten refer to the same thing. 

- Line 65: What do you mean by electrodeposition being a low energy consumption technology? Only compared to other processes with higher energy consumption? This is because, by itself, electrodeposition is a process that consumes large amounts of energy. 

- Line 74: please check the word "which", because possibly it should be changed to "with". 

- Line 80: please check the name of the system. Aluminum should be added?

- Line 111: please check the sentence: An aluminum wire (...) was "soaked" for one week to remove water. Was the cable soaked in the ionic solution to remove the water? What water? ambient humidity?

- How were the reactions 1 to 4 that are proposed to occur in the different parts of the electrochemical system characterized? Any reference at least?

- Lines 119 and 120: Is it correct that the liquids (distilled water, NaOH solution, and acetone) were ultrasonically cleaned to remove "surface impurities" and water? Please check this idea. 

- Line 140: please check the value of the Cu K(alpha) wavelength.  

-Line 152: Please omit the word And from And also...

Figure 3: you can omit the vertical axis label as it represents arbitrary and relative units. You can shorten the figure caption by changing it to XRD analyses of samples in Table 2. 

-Line 165: the effect of the current density on the value of the peak (111) is not clear. Please check this idea. 

- Line 190: please check the term "interval time". 

- Line 227: please the number of Figures 4(d), (e), and (f). They should be 5(d), 5(e), and 5(f). 

- Line 245: please check and avoid terms like we believe as they do not contribute much to the scientific analysis of your data. 

- Line 254:  It was missing to specify the operational variable that affected the quality of the deposited films. 

In the conclusions section, the contribution of the study and the potential of the process were not specified. 

Reviewer 2 Report

Review on "Aluminium Electrodeposition on the Surface of Boron Carbide Ceramics by Use EMIC-AlCl3 Ions Liquid"

1. Although the introduction is reasonable, the references are a bit old. There are just two papers from the last 5 years. Most of the cited work is from between 2000 - 2018, thus it feels that recent findings on the investigated topic were not discussed. Please improve.

2. Material and methods section presents the methodology used well. Please indicate the actual loading used during hardness measurements.

2.1 How the parameters used in Table 2 were chosen? Were they optimized somehow?

3. Results section needs to be improved significantly.

3.1 Based on Figure 2, please discuss how experimental conditions affect the shape of particles and what mechanisms stand behind them. Please relate to existing literature.

3.2 Please quantify and measure the particle size using average size/diameter. The relation between current density/frequency and an average particle size could be presented in the form of a graph to clearly expose the dependence.

3.3 XRD analysis: line 166 - "(...) the surface was the most obtained." - please rewrite, it is hard to understand.

3.4 Table 3: how crystal size and thickness were measured?

3.5 Table 3: confidence intervals are missing.

3.6 Line 183: "when current densities decreased, crystal size growth" - could it be explained? What is the reason?

3.7 Lines 187 - 188: "high frequency reduces the grain size" - could it be explained? What is the reason?

3.8 Figure 4: 5(c-1) and 6(d-1) - scale bars are missing

3.9 Line 224: "after the three-point bending test the aluminium film did not break" - did the authors perform three-point bending tests? The results were not shown. Furthermore, the SEM image or references should be provided to support such a statement. Otherwise, it cannot be claimed.

3.10 Line227 - "Figures 4d, e, and f" - please doublecheck. There is no such figure.

3.11 Lines 229 - 231: "Britlle phases are always (...) and hardening effect" - such statement needs to be supported by reference or SEM image.

3.12 Lines 231 - 235: "The surface hardness value of smaller grains is higher" - the hardness measurements for each specimen should be presented in the table.

3.13 Figure 5: scale bars are missing

4. Conclusions

4.1 Line 254: "the quality of the deposited films was affected." - by what the quality was affected? Be precise.

4.2 Line 259: "The frequency change had little effect..." - what is the reason?

The manuscript submitted needs to be significantly improved before further consideration for publication elsewhere. Please avoid the usage of "smaller", "lower" etc during comparison. Quantify where possible so the reader could clearly see the dependence. Please discuss the findings in relation to the published literature.

Decision: The manuscript will be reconsidered for publication after major revision.

Round 2

Reviewer 1 Report

Dear Authors: 

Although you took into account and made changes based on the comments made to the previous version of your manuscript, there are some important aspects that could be improved in terms of greater detail in the methodology, results, and conclusions sections. In this sense, I suggest you be clearer in the following:

- Regarding the selection of the levels of the test variables. 

- Taking into account that your conclusions are based almost exclusively on the measurement of deposit characteristics such as hardness, grain size, and film thickness, the values presented in Table 3 are single average values (measurements made on a single sample) or do they result from the average of independent measurements? Do you have some values of standard deviation or some range of variation?

- In this version of the manuscript, Figure 6 was used only to refer to a single occasion when there was the detachment of the film. I consider that it is necessary to strengthen the objective of including this figure in the manuscript. 

- You should still improve the conclusions section as you are only based on specific observations regarding the results without discussing the potential of the technique. 

This was requested in the last version. I consider that the changes made were for the benefit of the manuscript. However, you could clarify what is requested on this occasion. 

Best regards. 

Reviewer 2 Report

The manuscript still needs major revision:

1.       Comment 2.1 was not addressed. “2.1 How the parameters used in Table 2 were chosen? Were they optimized somehow?” Please explain in the manuscript.

2.       Comment 3.2 was not addressed. “3.2 Please quantify and measure the particle size using average size/diameter. The relation between current density/frequency and an average particle size could be presented in the form of a graph to clearly expose the dependence.

3.       Comment 3.9 was not addressed. “3.9 Line 224: "after the three-point bending test the aluminium film did not break" - did the authors perform three-point bending tests? The results were not shown. Furthermore, the SEM image or references should be provided to support such a statement. Otherwise, it cannot be claimed.

4.       Comment 4.2 was not addressed. “4.2 Line 259: "The frequency change had little effect..." - what is the reason?

5.       Table 3: The authors have shown average values of hardness and thickness. The reviewer believes that these results were obtained as average values from at least 3 independent measurements thus +/- ____ value should appear (for example 45HV±2, or 30µm±4)

6.       Figure 5: The SEM technique, or EDS in particular, is not an effective technique to evaluate oxygen distribution.

7.       Figure 6 is not clear at all. What is the point of presenting such a figure? The fracture surface observations should be performed. Furthermore, the conditions for three-point bending testing nor results were not presented.

Round 3

Reviewer 2 Report

The authors have improved the manuscript according to suggestions thus it could be considered for publication.

In order to improve the quality of the paper, the authors could include a short description of how the parameters used in Table 2 were determined.

Additionally, a short note about qualitative assessment of oxygen distribution using EDS should be added.
